# Structure-guided isoform identification for the human transcriptome

**Markus J Sommer[1,2]\*, Sooyoung Cha[3,4], Ales Varabyou[2,5], Natalia Rincon[1,2], Sukhwan Park[3,4], Ilia Minkin[1,2], Mihaela Pertea[1,2], Martin Steinegger[3,4,6]\*†, Steven L Salzberg[1,2,5,7]\*†**

[1]Department of Biomedical Engineering, Johns Hopkins School of Medicine and Whiting School of Engineering, Baltimore, United States; [2]Center for Computational Biology, Johns Hopkins University, Baltimore, United States; [3]School of Biological Sciences, Seoul National University, Seoul, Republic of Korea; [4]Artificial Intelligence Institute, Seoul National University, Seoul, Republic of Korea; [5]Department of Computer Science, Johns Hopkins University, Baltimore, United States; [6]Institute of Molecular Biology and Genetics, Seoul National University, Seoul, Republic of Korea; [7]Department of Biostatistics, Johns Hopkins University, Baltimore, United States

**\*For correspondence:**
markusjsommer@gmail.com
(MJS);
martin.steinegger@snu.ac.kr
(MS);
salzberg@jhu.edu (SLS)

†These authors contributed
equally to this work

**Competing interest:** The authors
declare that no competing
interests exist.

**Reviewing Editor:** Volker
Dötsch, Goethe University,
Germany

**Abstract** Recently developed methods to predict three-dimensional protein structure with high accuracy have opened new avenues for genome and proteome research. We explore a new hypothesis in genome annotation, namely whether computationally predicted structures can help to identify which of multiple possible gene isoforms represents a functional protein product. Guided by protein structure predictions, we evaluated over 230,000 isoforms of human protein-coding genes assembled from over 10,000 RNA sequencing experiments across many human tissues. From this set of assembled transcripts, we identified hundreds of isoforms with more confidently predicted structure and potentially superior function in comparison to canonical isoforms in the latest human gene database. We illustrate our new method with examples where structure provides a guide to function in combination with expression and evolutionary evidence. Additionally, we provide the complete set of structures as a resource to better understand the function of human genes and their isoforms. These results demonstrate the promise of protein structure prediction as a genome annotation tool, allowing us to refine even the most highly curated catalog of human proteins. More generally we demonstrate a practical, structure-guided approach that can be used to enhance the annotation of any genome.

## Editor's evaluation

This study applies AlphaFold to the CHESS selection of transcripts with the goal of generating predicted 3D protein structures and a quality measure of folding, the pLDDT score. From these data, the authors build a database for result exploration, documented by several examples, including proteins, where the authors propose the pLDDT score as a measure of presumed superior biological functionality over other isoforms. These results will be highly relevant for anyone working with proteins that occur in different isoforms.

## Introduction

More than 20 years after the initial publication of the human genome, the scientific community is still trying to determine the complete set of human protein-coding genes. Although the number of genes is converging around 20,000, we do not yet have agreement on the precise number. The true number

of different isoforms of human genes – variations due to alternative splicing, alternative transcription initiation sites, and alternative transcription termination sites – is even less certain. Currently, the major human gene annotation databases each contain well over 100,000 protein-coding transcripts (*Howe et al., 2021*; *O'Leary et al., 2016*; *Harrow et al., 2012*; *Pertea et al., 2018*; *Salzberg, 2018*), but the sets of transcripts vary widely among them. The disagreement between human transcriptome databases was clearly demonstrated when, in 2018, GENCODE and RefSeq were shown to agree on fewer than 50,000 of the nearly 300,000 total transcripts in their human annotations (*Pertea et al., 2018*).

Although the functions of many human genes are known, elucidating gene function remains a complex and time-consuming task. Given that at least 92% of human genes express more than one isoform (*Wang et al., 2008*), and that the human transcriptome contains an average of seven or more unique transcripts per protein-coding gene (*Tung et al., 2020*), the only feasible way to determine which isoforms are functional on a genome-wide scale is by using computational methods. Until now, the primary tools used to investigate gene function were sequence alignment and gene expression. Alignment relies on the long-established observation that if a protein is conserved in other species, then it is likely to be functional, particularly if the conservation extends to distantly related species (*Lindblad-Toh, 2011*). This rule applies to isoforms as well: if we can find evidence that a particular sequence – e.g., a protein that uses an alternative exon – is present in species that diverged tens of millions of years ago, then the conservation of the sequence argues in favor of its function.

In a similar vein, the use of RNA sequencing (RNA-seq) to detect gene expression also provides clues to function: if a transcript is consistently expressed in multiple samples, it is more likely to be functional than one for which little expression evidence can be found. Genes may encode multiple transcripts that fold into distinct isoforms with well-defined functions (*Wang et al., 2008*), but recent work has shown that most assembled human transcripts are found at very low levels in the transcriptomes of individual tissues (*Pertea et al., 2018*), and may simply reflect biological noise, products of intrinsically stochastic biochemical reactions (*Eling et al., 2019*; *Ponting and Haerty, 2022*). The large majority of assembled isoforms are unlikely to be functional, and indeed only a small percentage are included in current human genome annotation databases (*Howe et al., 2021*; *O'Leary et al., 2016*; *Harrow et al., 2012*; *Pertea et al., 2018*; *Salzberg, 2018*). Because transcription is noisy, the observation of transcription in RNA-seq data is insufficient evidence to conclude that a sequence is functional (*Palazzo and Lee, 2015*).

This study explores a fundamentally new line of evidence that can be used to investigate protein function: computational prediction of three-dimensional (3D) structure. The recently developed AlphaFold2 system can automatically predict 3D protein structure with accuracy that often matches far more time-consuming laboratory methods (*Jumper et al., 2021*; *Tunyasuvunakool et al., 2021*), allowing us to generate structure predictions for thousands of gene isoforms. In proteins where a substantial portion folds into an ordered structure, estimated to be 68% of human proteins (*Tunyasuvunakool et al., 2021*; *Deiana et al., 2019*), a well-folded structure within an isoform argues in favor of its functionality. Conversely, a poorly folded isoform may indicate loss of function.

In a recent effort to create a single consensus annotation of all human protein-coding genes, two of the leading human genome annotation centers created the MANE (Matched Annotation from NCBI and EMBL-EBI) database (*Morales et al., 2022*), a high-quality collection of protein-coding isoforms for which the annotation databases RefSeq (NCBI) and Ensembl-GENCODE (EMBL) match precisely. The goal of MANE is to identify just one isoform for each protein-coding gene that is well supported by experimental data, and to ensure that both databases agree on all exon boundaries as well as the sequence of the associated protein. In addition to the one-isoform-per-gene collection known as MANE Select, a small number of additional transcripts with special clinical significance, known as MANE Plus Clinical, are included in the database. Upon its initial release, MANE included only around 50% of human protein-coding genes. The latest version, v1.0, includes 19,062 genes and 19,120 transcripts, with an additional 58 transcripts included in the MANE Plus Clinical set. These transcripts have been described as a 'universal standard' for human gene annotation, and they provide a valuable resource to scientists and clinicians who need a consistent set of functional primary transcripts.

Here, we describe our use of protein folding predictions from ColabFold (*Mirdita et al., 2022*), an open-source accelerated version of AlphaFold2, alongside experimental RNA-seq expression data from the Genotype-Tissue Expression (GTEx) project (*GTEx Consortium, 2013*), to present substantial evidence for functional isoforms that can be used to improve human gene annotation, including the

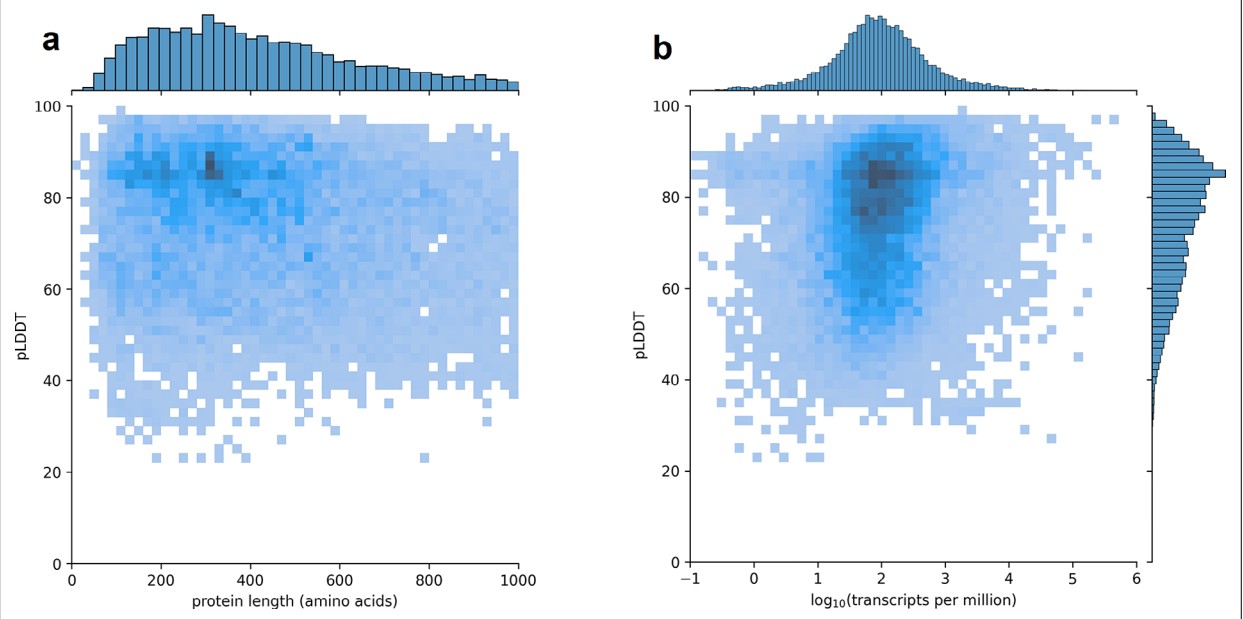

**Figure 1.** Predicted local distance difference test (pLDDT) distribution across the human transcriptome. Two-dimensional joint histograms comparing pLDDT to protein amino acid length (**a**) and expression (**b**) measured in transcripts per million (TPM). For each protein-coding gene, only the isoform found in the highest number of Genotype-Tissue Expression (GTEx) samples is plotted. No strong trend is visible in the relationship between pLDDT and either protein length (**a**) or transcript expression (**b**).

MANE gene set as well as the comprehensive human annotation databases RefSeq (*O'Leary et al., 2016*), GENCODE (*Harrow et al., 2012*), and the Comprehensive Human Expressed SequenceS (CHESS) database (*Pertea et al., 2018*). We dive into a few exemplary predictions to explain, biologically and evolutionarily, the 3D structure of our alternate isoforms. We also present an example of a novel protein isoform in mouse to demonstrate the general applicability of this structure-guided approach to improving functional annotation of any genome.

## Results

### Scoring the transcriptome

Using CHESS, a large set of transcripts assembled from nearly 10,000 human RNA-seq experiments, we identified all protein-coding gene isoforms that were 1000aa or less in length (see Materials and methods). The 233,973 transcripts at 20,666 gene loci that fit this description encoded 127,398 distinct protein isoforms, and we predicted structures for all of them. Additionally, we included 3302 structure predictions for proteins with length >1000aa from the AlphaFold Protein Structure Database (*Varadi et al., 2022*) for isoforms with an exact protein sequence match in CHESS 3. This resulted in a total of 237,295 transcripts at 20,817 gene loci encoding 130,700 distinct protein isoforms. As shown in *Figure 1*, we observe no strong trend in the relationship between predicted local distance difference test (pLDDT) and either protein length or overall transcript expression in the GTEx data. The lack of a clear linear relationship implies protein structure prediction may provide an orthogonal source of useful information for genome annotation efforts.

In total, 22,644 transcripts (9.7% of all examined transcripts) encoded an isoform that scored a higher pLDDT than the isoform encoded by the corresponding MANE transcript. However, many of these higher-scoring transcripts encoded relatively short, often low-expressed, likely non-functional fragments of larger proteins. Therefore, we incorporated filters based on RNA-seq expression data when determining which higher-scoring isoforms appeared superior to MANE isoforms (see 'Filtering MANE comparisons'). Based on the combination of RNA-seq evidence and protein foldability, we identified 940 unique alternate isoforms at 632 loci (3.4% of all MANE loci) which appeared to have a more stable structure than the annotated primary isoform. Data for these 940 alternate isoforms can be found in *Supplementary file 3*. Worth noting is that over 96% of the MANE loci evaluated here

contained no higher-scoring alternate isoforms that passed our filtering criteria. This is a testament to both the high degree of consistency in MANE and the sensitivity of protein structure prediction for finding instances where alternative isoforms may create functional products. Additionally, for 35% of all human protein-coding gene loci in our analysis, the most commonly observed isoform scored a pLDDT below 70, suggesting that intrinsic disorder may be an important feature of proteins at these loci.

Gene identifiers for all predicted protein isoforms as well as pLDDT scores and evolutionary conservation data from mouse can be found in *Supplementary file 1*. Predicted scores and GTEx expression data for all isoforms overlapping a MANE locus can be found in *Supplementary file 2*. All predicted protein structures as well as data for all tables in this paper are publicly available at our website, isoform.io.

## Exemplary predictions

To illustrate the improvements in human gene annotation that can be obtained using accurate structure prediction, we describe a small set of proteins, selected from *Supplementary file 3*, where an alternate isoform appears to be superior to the isoform chosen for inclusion in MANE. For these examples, the alternative isoform is clearly functional based on structure as well as evolutionary conservation

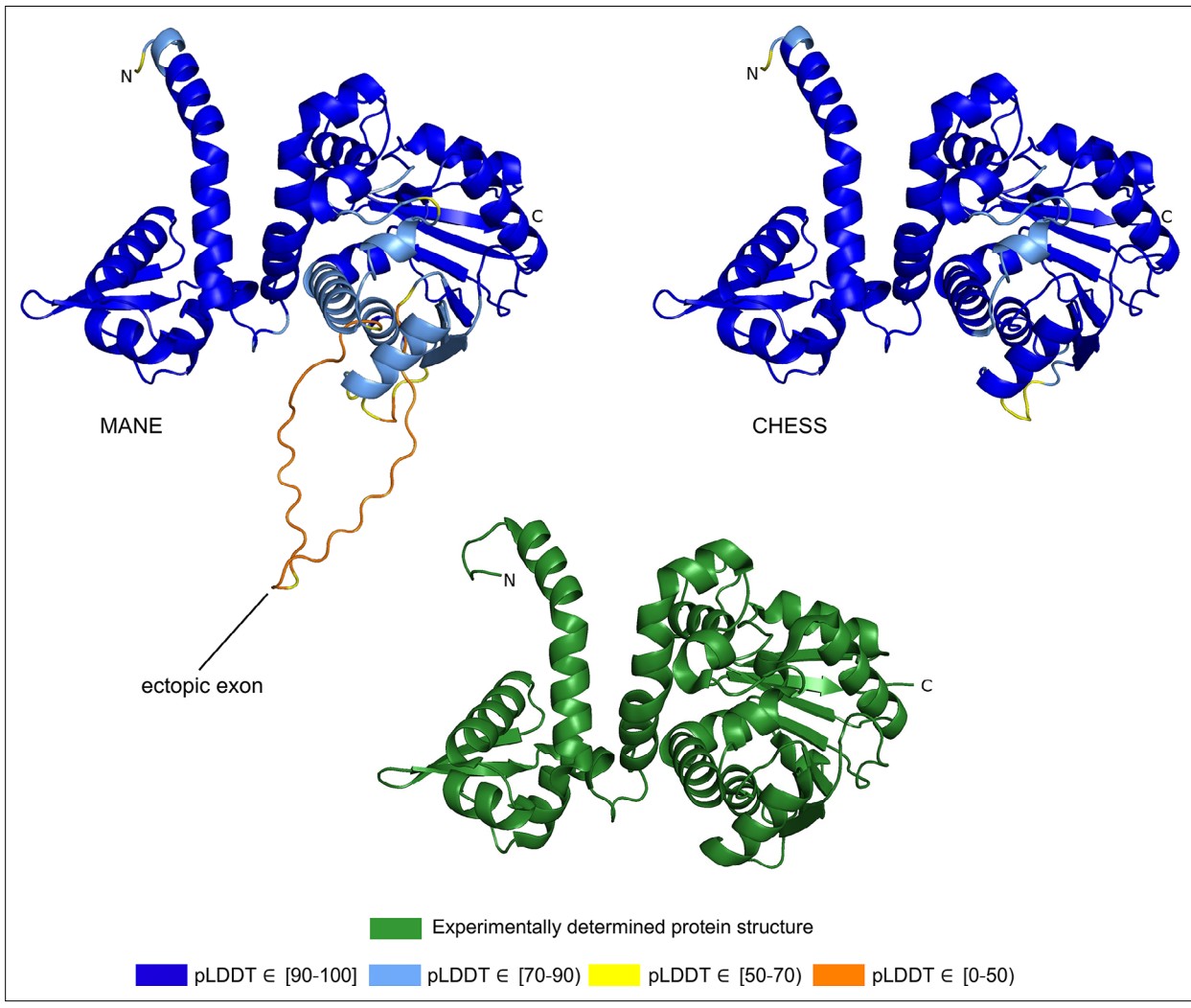

**Figure 2.** Acetylserotonin *O*-methyltransferase (ASMT) isoform comparison. Comparison of predicted structures of ASMT, showing the 373aa isoform from Matched Annotation from NCBI and EMBL-EBI (MANE) (CHS.57426.2, RefSeq NM_001171038.2, GENCODE ENST00000381241.9) on the left, and a 345aa alternate isoform from Comprehensive Human Expressed SequenceS (CHESS) (CHS.57426.4, GENCODE ENST00000381229.9) on the right. The CHESS 345aa isoform closely matches the experimentally determined X-ray crystal structure of the biologically active protein (*Botros et al., 2013*), shown at the bottom.

and, in some cases, additional expression evidence from RNA-seq data. For some of these examples, the MANE isoform is missing critical structural elements and may not be functional at all.

## Acetylserotonin *O*-methyltransferase

Acetylserotonin *O*-methyltransferase (*ASMT*, alternatively *HIOMT*) is responsible for the final catalytic step in the production of melatonin, a critical hormone in sleep, metabolism, immune response, and neuronal development (*Melke et al., 2008*). Depressed levels of circulating melatonin have been associated with autism spectrum disorder, and clinical studies have classified *ASMT* as a susceptibility gene due to the highly significant association between *ASMT* activity and autism (*Melke et al., 2008*; *Rossignol and Frye, 2011*).

The CHESS and GENCODE gene databases contain a 345aa isoform of *ASMT* (CHS.57426.4, ENST00000381229.9) while RefSeq is missing this isoform. The MANE version of this gene is 373aa long and appears in the CHESS (CHS.57426.2), GENCODE (ENST00000381241.9), and RefSeq (NM_001171038.2) gene databases. The predicted structures of both isoforms are shown in *Figure 2*.

We hypothesized that the highest scoring isoform of *ASMT* according to AlphaFold2 corresponds to the biologically active version of the protein. The score we use for these comparisons is the pLDDT score, which has been demonstrated to be a well-calibrated, consistent measure of protein structure prediction accuracy (*Jumper et al., 2021*; *Tunyasuvunakool et al., 2021*). A pLDDT score above 70 (the maximum is 100) indicates that a predicted structure can generally be trusted, while a score below 70 may indicate folding prediction failure or intrinsic disorder within a protein. Scores above 90 imply structure predictions accurate enough for highly shape-sensitive tasks such as chemical binding site characterization. The structure of the 345aa isoform has a very high pLDDT score of 94.7, versus the somewhat lower score of 87.1 for the 373aa MANE isoform.

Because melatonin is primarily synthesized within the human pineal gland at night, we quantified ASMT isoform expression using RNA-seq data from a previously published experiment that used tissue extracted from the pineal gland of a patient who died at midnight (*Chang et al., 2020*). In this tissue sample, the 345aa isoform of *ASMT* was expressed at a level of 327 transcripts per million (TPM), while the 373aa isoform from MANE was expressed at 34 TPM, nearly 10 times lower, supporting our hypothesis that the higher scoring 345aa isoform is functional.

Further evidence for the functionality of the 343aa isoform is shown in *Figure 2*. An ectopic exon in the MANE *ASMT* protein creates an unstructured loop that bulges out from the primary structure. The alternate isoform, missing this ectopic exon, closely matches the experimentally determined *ASMT* X-ray crystal structure of the biologically active protein. Furthermore, as reported in *Botros et al., 2013*, the insertion of exon 6, corresponding to the ectopic exon in the MANE isoform, distorts the structure and destroys its ability to bind *S*-adenosyl-L-methionine and to synthesize melatonin. Thus, the structural comparison, the expression evidence, and a melatonin synthesis activity assay all combine to support our hypothesis that the 345aa isoform represents the primary biologically functional isoform of *ASMT*.

## Gamma-N crystallin

Gamma-N crystallin (*CRYGN*) is a highly conserved member of the crystallin family of proteins, responsible for the transparency of the lens and cornea in vertebrate eyes (*Andley, 2007*). The intron-exon structure of *CRYGN* has been conserved across at least 400 million years of vertebrate evolution, with close orthologs present in the genomes of chimpanzees, mice, frogs, and the white-rumped snowfinch. Given this extensive evolutionary history, *Wistow et al., 2005*, were surprised to observe that the primate *CRYGN* gene has lost its canonical stop codon, leading them to conclude 'the human gene has clearly changed its expression and may indeed be heading for extinction'.

As shown in *Figure 3a*, the MANE isoform (CHS.52273.5, RefSeq NM_144727.3, GENCODE ENST00000337323.3) that matches descriptions by *Wistow et al., 2005*, includes sequences that do not fold well, as indicated by its pLDDT score of 67.7. However, we found an alternate *CRYGN* isoform, assembled from *GTEx Consortium, 2013* data, that had a far higher pLDDT score of 92.2, shown in *Figure 3b*. Small differences between pLDDT scores may not be meaningful, but large score differences, such as the 24-point gap between the two isoforms of *CRYGN* discussed here, represent a substantial difference in prediction confidence across a large portion of the protein. The higher-scoring

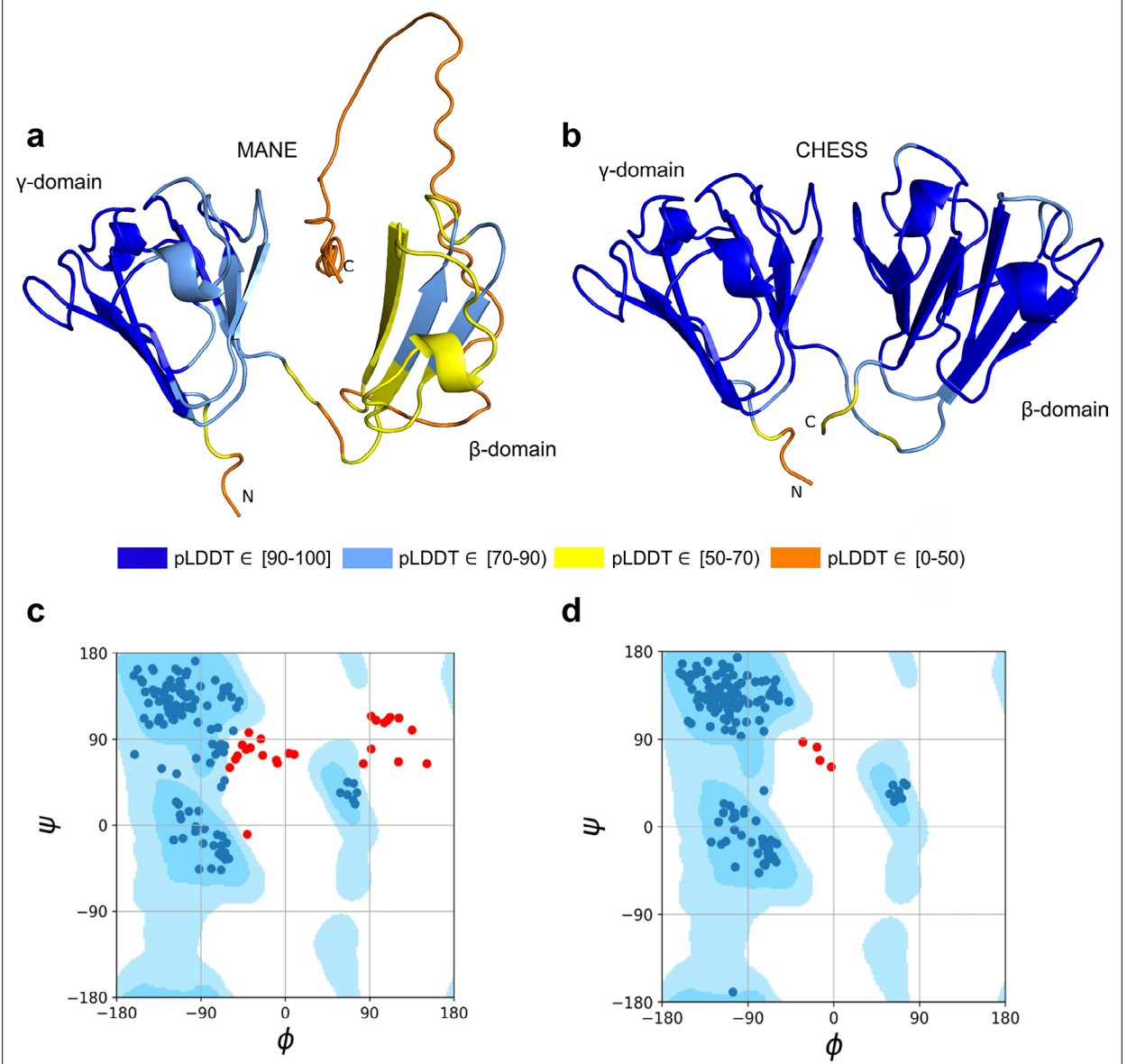

**Figure 3.** CRYGN isoform comparison. (**a**) Predicted protein structure for the Matched Annotation from NCBI and EMBL-EBI (MANE) isoform (CHS.52273.5, RefSeq NM_144727.3, GENCODE ENST00000337323.3) of gamma-N crystallin (CRYGN), colored by predicted local distance difference test (pLDDT) score. (**b**) Predicted protein structure for a CRYGN alternate isoform (CHS.52273.9, GENCODE ENST00000644350.1). (**c**) Ramachandran plot for the MANE (CRYGN) isoform. Dark blue areas represent 'favored' regions while light blue represent 'allowed' regions (*Lovell et al., 2003*). The 32 red dots represent amino acid residues with secondary structures that fall outside the allowed regions. (**d**) Ramachandran plot for the alternate CRYGN isoform with 4 red dots falling in disallowed regions, compared to 32 disallowed in MANE. All residues associated with the 4 red dots in the alternate isoform are shared with the MANE isoform in the poorly folded N-terminal region.

*CRYGN* isoform is present in CHESS (CHS.52273.9) and GENCODE (ENST00000644350.1), and it was also present in RefSeq v109 (XM_005249952.4) but was removed in the next release, v110.

Both the MANE and alternate isoforms are exactly the same length despite having different C-terminal sequence content. Visual comparison of the predicted structure for the alternate isoform reveals a marked improvement in the structure in the β domain and a clear recovery of *CRYGN*'s dimer-like characteristic, with two structurally similar domains as shown in *Figure 3b* and *Video 1*. Ramachandran plots (*Figure 3c and d*) also support the structure of the alternate CHESS isoform.

Encouraged by the substantially improved folding of this alternate isoform, we examined the intron-exon structure of *CRYGN* in human to determine how it recovered its functional shape despite

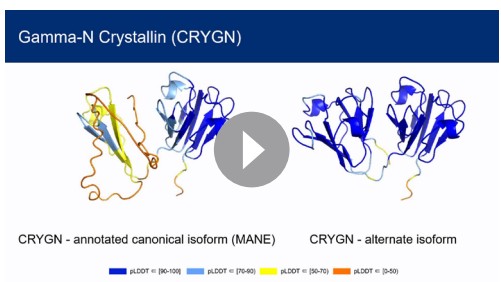

**Video 1.** CRYGN comparison. A three-dimensional (3D) animation comparing the predicted protein structure of the Matched Annotation from NCBI and EMBL-EBI (MANE) isoform (CHS.52273.5, RefSeq NM_144727.3, GENCODE ENST00000337323.3) of gamma-N crystallin (CRYGN) versus the predicted protein structure for the highest-scoring CRYGN alternate isoform (CHS.52273.9, GENCODE ENST00000644350.1). https://elifesciences.org/articles/82556/figures#video1

losing its original stop codon. We found that the common vertebrate four-exon structure of *CRYGN* has changed to a five-exon structure in humans, as shown in *Figure 4*. In the well-folded alternate isoform, a novel primate-specific splice site removes the last four amino acids as compared to other vertebrates, but the new primate-specific fifth exon contains a downstream stop codon that adds four residues. The poorly folding MANE isoform (*Figure 4*, bottom), in contrast, entirely skips the fourth exon, resulting in a frameshift that adds 43 C-terminal amino acids which have no similarity to any *CRYGN* sequence outside of primates.

## Thioredoxin domain-containing protein 8

The thioredoxin protein family represents an ancient group of highly conserved small globular proteins found in all forms of life (*Modi et al., 2018*). Thioredoxin domain-containing protein 8 (*TXNDC8*, alternatively *PTRX3*) is a testis-specific enzyme responsible for catalyzing redox reactions via the oxidation of cysteine from dithiol to disulfide forms (*Jiménez et al., 2004*).

As shown in *Figure 5*, several canonical protein motifs appear altered or missing in the predicted structure of the human *TXNDC8* MANE transcript, as it lacks a highly conserved sequence that should start only eight residues away from the CGPC dithiol/disulfide active site. The α2 helix is severely truncated, leading directly to the α3 helix, thereby entirely skipping the β3 sheet. The β5 sheet, normally providing an interaction bridge between the α3 and α4 helices, is similarly missing in its entirety. Finally, the α4 helix is present but rotated 140 degrees relative to its canonical position. These large alterations to the fundamental thioredoxin protein structure result in the MANE transcript receiving a pLDDT score of 56.7.

In stark contrast to the poorly folded MANE isoform (CHS.56446.8, RefSeq NM_001286946.2, GENCODE ENST00000423740.7), an alternate isoform assembled as part of the CHESS project (CHS.56446.14) and RefSeq (NM_001364963.2) has a pLDDT score of 96.9, an improvement of 40 points. Inspection of the alternate transcript (*Figure 5*) reveals full recovery of the canonical α2, α3,

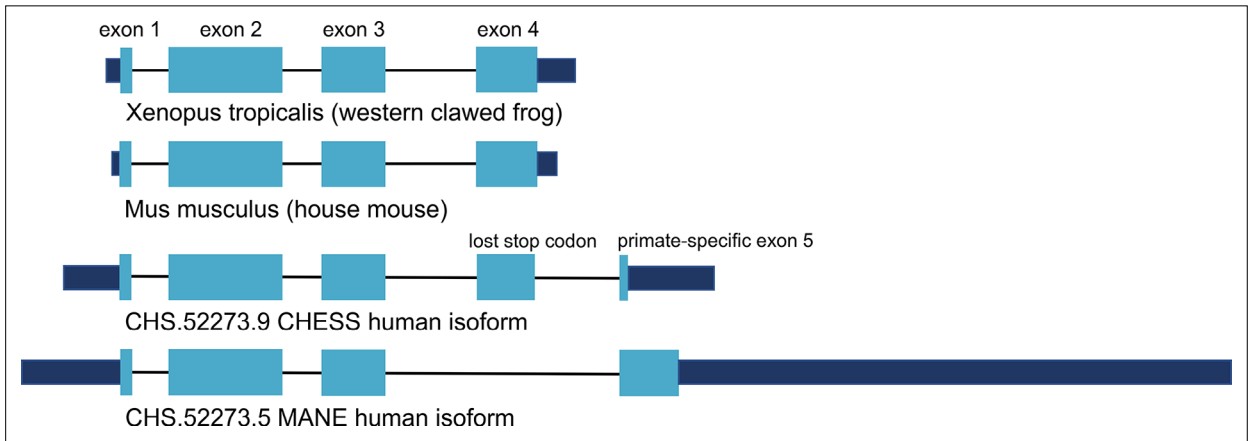

**Figure 4.** CRYGN intron-exon structure. Comparison of gamma-N crystallin (CRYGN) transcript structures in frog, mouse, and human. Exons 1, 2, and 3 are highly conserved across all species. Exon 4 is missing from the poorly folding Matched Annotation from NCBI and EMBL-EBI (MANE) isoform, while exon 5 shows no homology to any species outside of primates. The loss of a stop codon in human exon 4 appears to be balanced by the inclusion of a short novel exon that adds only four amino acids to the final protein. Coding portions of exons are shown with thicker rectangles in teal. Intron lengths are reduced proportionately for the purpose of display.

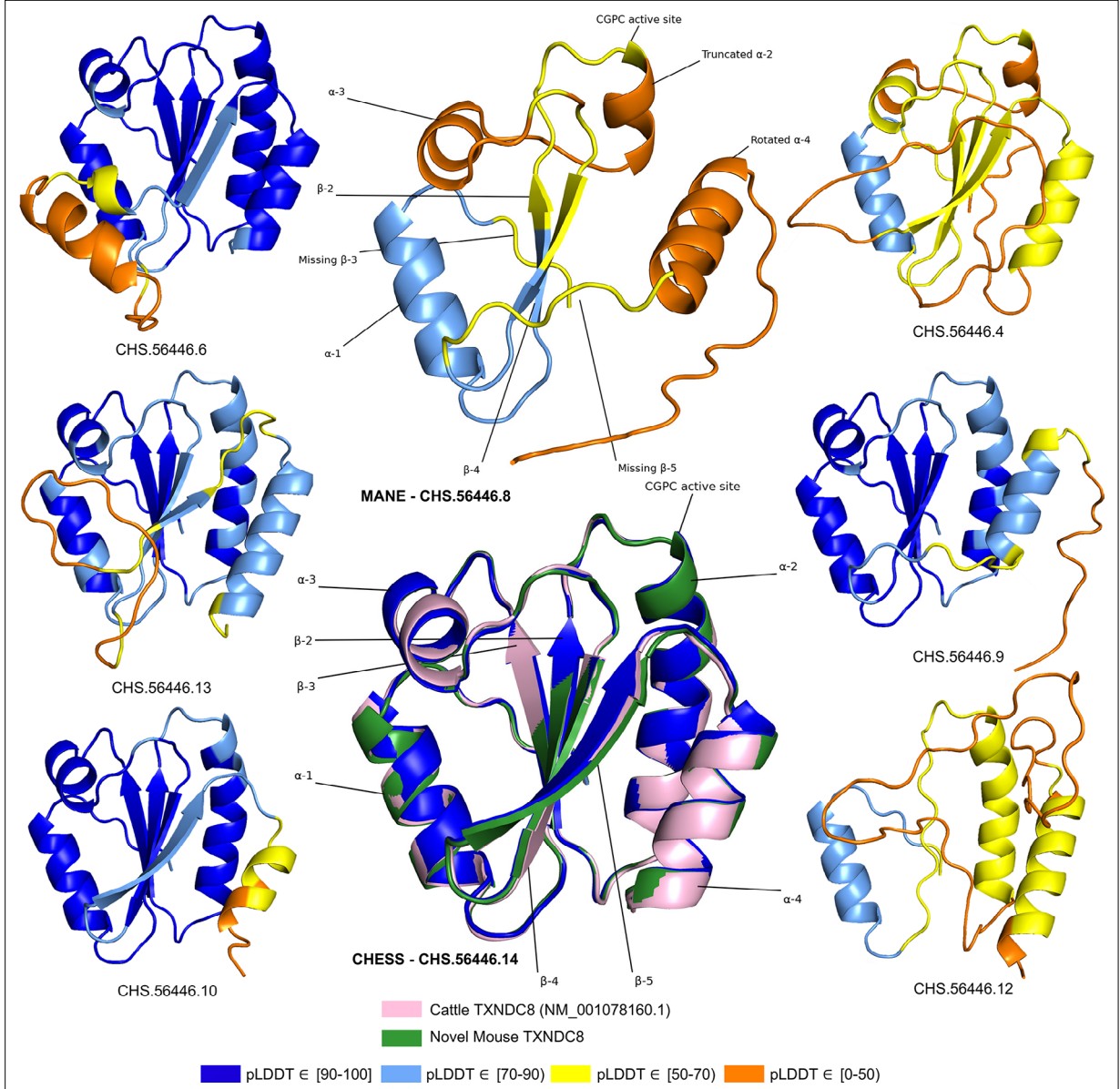

**Figure 5.** TXNDC8 isoform comparison. Predicted protein structures for seven distinct human isoforms of thioredoxin domain-containing protein 8 (TXNDC8), as well as the primary cattle transcript and a novel mouse transcript. Alternate human isoforms 4, 9, and 12 (right side of figure) lack multiple canonical thioredoxin structures and thus appear non-functional. Several canonical protein motifs are missing or altered in the predicted structure of the Matched Annotation from NCBI and EMBL-EBI (MANE) transcript (top center). In contrast, the alternate human transcript 14 matches cattle and mouse to within 0.8 Å. Human transcript CHS.56446.14 is colored solid dark blue because every amino acid residue scores a predicted local distance difference test (pLDDT) above 90.

and α4 helices as well as the β3 and β5 sheets. Moreover, 3D alignment of the protein encoded by CHS.56446.14 to the protein from the primary *TXNDC8* isoform in *Bos taurus* (cow) reveals a very close structural correspondence between the two proteins, with a predicted root-mean-square deviation (RMSD) of 0.8 Å. *Figure 5* shows multiple alternative isoforms of human *TXNDC8* from the CHESS annotation, as well as the 3D alignment of CHS.56446.14 to its orthologs in cow and mouse. Given its substantially higher pLDDT score and near-perfect structural conservation in other species, the CHESS transcript appears to be a much better candidate for the canonical form of this protein. The MANE isoform, because it is missing multiple key structures, may represent a non-functional product of transcriptional noise.

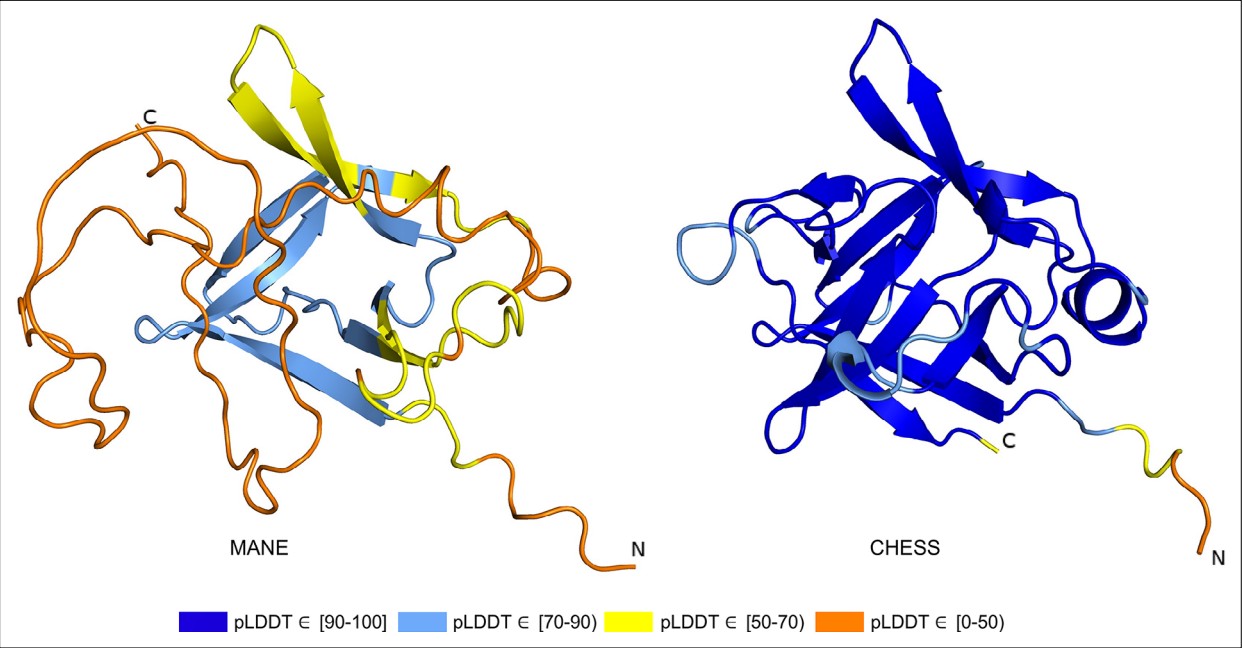

**Figure 6.** IL36B isoform comparison. Comparison of predicted structures for interleukin 36 beta (IL36B) for the Matched Annotation from NCBI and EMBL-EBI (MANE) isoform (CHS.30565.1, RefSeq NM_014438.5, GENCODE ENST00000259213.9) and an alternate isoform from Comprehensive Human Expressed SequenceS (CHESS) and RefSeq (CHS.30565.4, RefSeq XM_011510962.1). The highly conserved protein sequence of the alternate human isoform achieves a very high predicted local distance difference test (pLDDT) score of 93.0, versus the MANE isoform's much lower pLDDT of 50.2.

All isoforms of *TXNDC8* shown in *Figure 5* were assembled from RNA-seq data during the construction of the CHESS database. This figure illustrates another potential use of structure prediction, namely the ability to distinguish among multiple functional and non-functional isoforms when annotating a genome. As discussed above, the MANE *TXNDC8* isoform appears non-functional, lacking several key structures. In addition, isoform 12 in *Figure 5* appears clearly non-functional, lacking all four of the β sheets and one of the α helices of isoform 14. Isoforms 4 and 9 also appear likely to be non-functional: both are missing one of the β sheets, and isoform 4 has a low pLDDT score of just 52. Although this example is only one of many, it illustrates how one can employ accurate 3D structure prediction in virtually any species as a powerful new tool to improve gene annotation.

Assemblies of RNA-seq experiments typically reveal thousands of un-annotated gene isoforms, as was illustrated by the use of GTEx to discover more than 100,000 new isoforms when building the original CHESS gene database (*Pertea et al., 2018*). The approach used here, computationally folding each distinct protein encoded by alternate isoforms, allows us to compare the structure of these predicted proteins to the 'best' structure for each protein-coding gene locus. For those proteins with at least one high-confidence structure, this strategy may allow us to identify and remove potentially non-functional isoforms.

## Interleukin 36 beta

Interleukin 36 beta (*IL36B*, alternatively *IL1F8*, *FIL1-ETA*, or *IL1H2*) mediates inflammation as part of a signaling system in epithelial tissue. The pro-inflammatory properties of *IL36B* have been implicated in the pathogenesis of psoriasis (*Carrier et al., 2011*), a common disease characterized by scaly rashes on the skin. In vitro and in vivo studies have found consistently increased expression of *IL36B* in psoriatic lesions, making the gene a potential target for future anti-psoriatic drugs (*Uppala et al., 2021*).

The ColabFold structure of the *IL36B* MANE isoform (CHS.30565.1, RefSeq NM_014438.5, GENCODE ENST00000259213.9) averaged a pLDDT of only 50.2, a score indicative of near-complete folding failure, while an alternate isoform in CHESS (CHS.30565.4) and RefSeq (XM_011510962.1), shown in *Figure 6*, scored 93.0, the largest relative score increase of any isoform we examined. This alternate isoform contains two C-terminal exons that are not present in the MANE isoform and that contribute nearly half of the total protein sequence. A BLASTP homology search of the 34aa coding

sequence of the final C-terminal exon in the MANE transcript yielded no hits beyond primates using default search parameters. In contrast, a BLASTP search of sequence unique to the alternate isoform revealed significant similarity (e-values of 0.001 and smaller) to *IL36B* orthologs in 1517 organisms including mouse, rat, and Hawaiian monk seal. In addition, expression of the MANE isoform was observed in only 1 sample in the GTEx data with an expression level of just 0.01 TPM, while the CHESS isoform was found in 775 samples with a much higher expression level of 8.4 TPM (*Supplementary file 3*).

Further probing the functionality of our high-scoring isoform, we aligned the predicted 3D structures for *IL36B* in human, mouse, and rat. As expected, the mouse and rat proteins aligned to each other remarkably well, with an RMSD of 0.60 Å. We found the low-scoring MANE protein aligned poorly to the structures for mouse and rat, averaging a distance of 2.74 Å, while the alternate isoform aligned far better with an RMSD of 0.76 Å. This close similarity in both sequence and structure to conserved orthologs in distant species strongly reinforces the argument that the alternate isoform represents the functional version of the protein in human.

## Post-GPI attachment to proteins 2

The protein known as post-GPI attachment to proteins 2 (*PGAP2*, alternatively *FRAG1* or *CWH43N*) is required for stable expression of glycosylphosphatidylinositol (GPI)-anchored proteins (*Tashima et al., 2006*) attached to the external cellular plasma membrane via a post-translational modification system ubiquitous in eukaryotes (*Englund, 1993*). Mutations in GPI pathway proteins have been linked to a wide variety of rare genetic disorders (*Bellai-Dussault et al., 2019*), while mutations in *PGAP2* specifically have been shown to cause to intellectual disability, hyperphosphatasia, and petit mal seizures (*Hansen et al., 2013*).

Out of 85 GTEx-assembled transcripts for *PGAP2* produced during the latest build of the CHESS database, encoding 33 distinct protein isoforms, the single highest scoring isoform according to ColabFold was CHS.7860.59 (RefSeq NM_001256240.2, GENCODE ENST00000463452.6), with a pLDDT of 87.9. The coding sequence of this isoform exactly matches the sequence of the assumed biologically active protein (*Hansen et al., 2013*; *Krawitz et al., 2013*), and all intron boundaries are conserved in mouse. For comparison, the annotated MANE protein (CHS.7860.58, RefSeq NM_014489.4, GENCODE ENST00000278243.9) has a pLDDT of 78.0 and the intron boundaries are not conserved in mouse. Predicted structures of both proteins are shown in *Figure 7*.

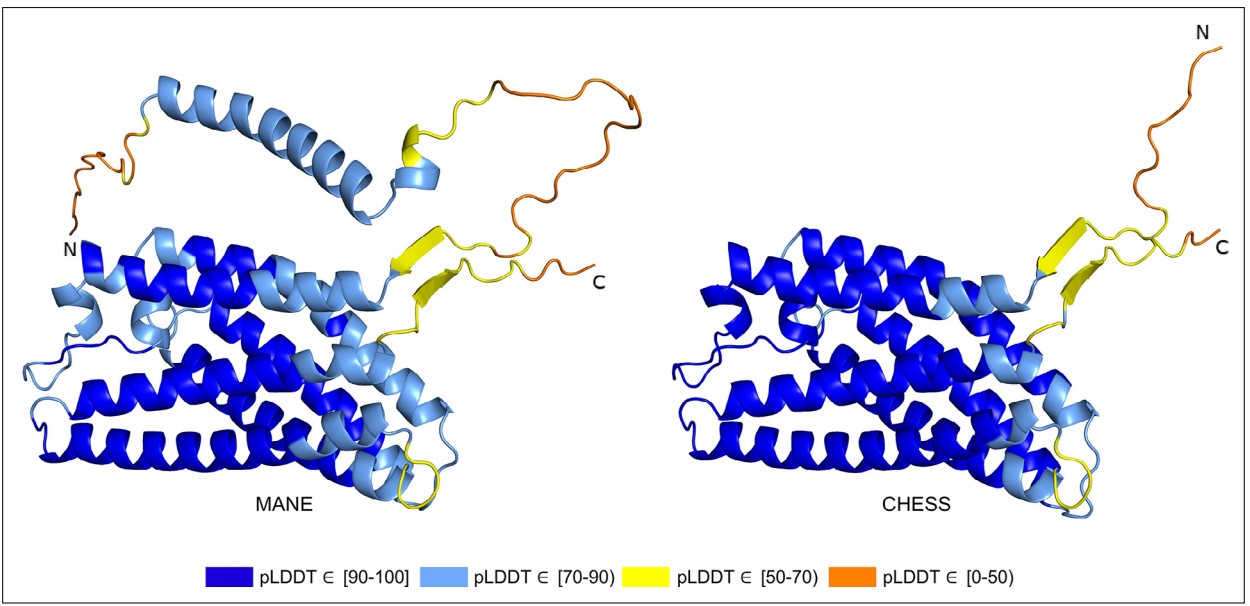

pLDDT ∈ [90-100]    pLDDT ∈ [70-90]    pLDDT ∈ [50-70]    pLDDT ∈ [0-50)

**Figure 7.** PGAP2 isoform comparison. Comparison of the structure of the Matched Annotation from NCBI and EMBL-EBI (MANE) isoform (CHS.7860.58, RefSeq NM_014489.4, GENCODE ENST00000278243.9) versus the highest scoring alternate isoform (CHS.7860.59, RefSeq NM_001256240.2, GENCODE ENST00000463452.6) for PGAP2. Of 33 distinct annotated protein isoforms of PGAP2, the one with the highest predicted local distance difference test (pLDDT) represents the biologically active version (*Hansen et al., 2013*; *Krawitz et al., 2013*) of PGAP2 in humans.

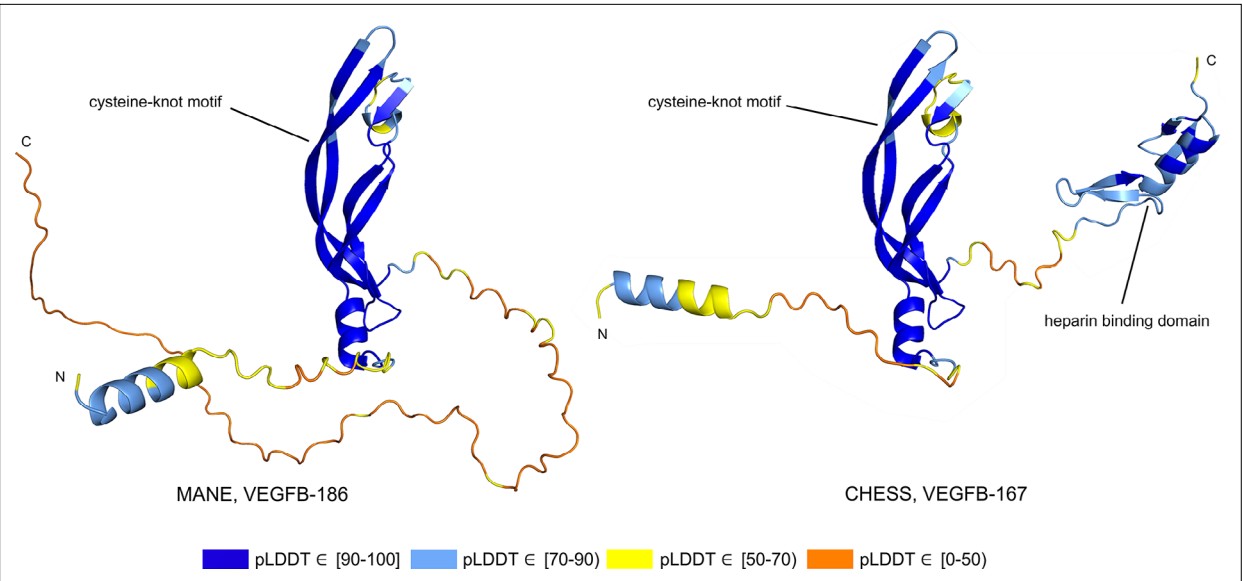

**Figure 8.** Vascular endothelial growth factor B (VEGFB) isoform comparison. VEGFB isoforms VEGFB-186 (**a**) and VEGFB-167 (**b**). The inclusion of a heparin binding domain in VEGFB-167 results in sequestration to the cell surface while VEGFB-186 remains freely soluble. Relying solely on predicted local distance difference test (pLDDT) comparisons in this case would be misleading, as both isoforms represent well-understood functional protein products.

RNA-seq data from GTEx also showed that the higher scoring isoform, CHS.7860.59, was expressed in 8776 samples with an average expression level of 2.6 TPM, compared to only 4116 samples with a 0.9 TPM average for the MANE isoform. Comparing their intron-exon structure revealed that the MANE transcript has one extra exon (the second exon out of six). On average, across all 31 tissues in the GTEx data, five times more spliced reads supported skipping that exon, as in CHS.7860.59, rather than including it.

## Functional splice variants may not fold well

Alternative splicing allows genes to code for multiple functional protein products (*Matlin et al., 2005*). Thus, rejecting all but the top-scoring isoform based on predicted structure may eliminate lower-scoring yet functional proteins. The risk of discarding functional transcripts by relying too heavily on the pLDDT score is well-illustrated by vascular endothelial growth factor B (*VEGFB*), a growth factor implicated in cancer and diabetes-related heart disease (*Lal et al., 2018*). The human *VEGFB* gene encodes two well-characterized protein isoforms: *VEGFB-167* and *VEGFB-186*. Alternative splicing that skips part of the sixth exon in *VEGFB-167* leads to sequestration of the protein to the cell surface due to a highly basic C-terminal heparin binding domain. Full inclusion of exon six in *VEGFB-186* results in a soluble protein freely transported to the blood stream (*Li, 2010*).

Both isoforms shown in *Figure 8* represent highly expressed and similarly functional products, containing a well-conserved cysteine-knot motif (*Iyer and Acharya, 2011*), yet *VEGFB-167* receives a pLDDT score of 81.7 while *VEGFB-186* receives a much lower pLDDT score of 69.5. The MANE isoform (CHS.9039.1, RefSeq NM_003377.5, GENCODE ENST00000309422.7) encodes the freely soluble protein *VEGFB-186*, while the alternate isoform (CHS.9039.2, RefSeq NM_001243733.2, GENCODE ENST00000426086.3) encodes the sequestered protein *VEGFB-167*. Additionally, *VEGFB-186* is present as a full-length cDNA clone (MGC:10373 IMAGE:4053976) in the Mammalian Gene Collection (*Temple et al., 2009*), a fact which strongly supports its functionality. Due to the large pLDDT score difference between the two functional *VEGFB* isoforms, a naïve attempt to use protein folding prediction scores as the sole oracle of protein function might inadvertently discard *VEGFB-186*, a clearly functional transcript. Thus, one must be careful to incorporate multiple sources of information when making decisions about isoform functionality.

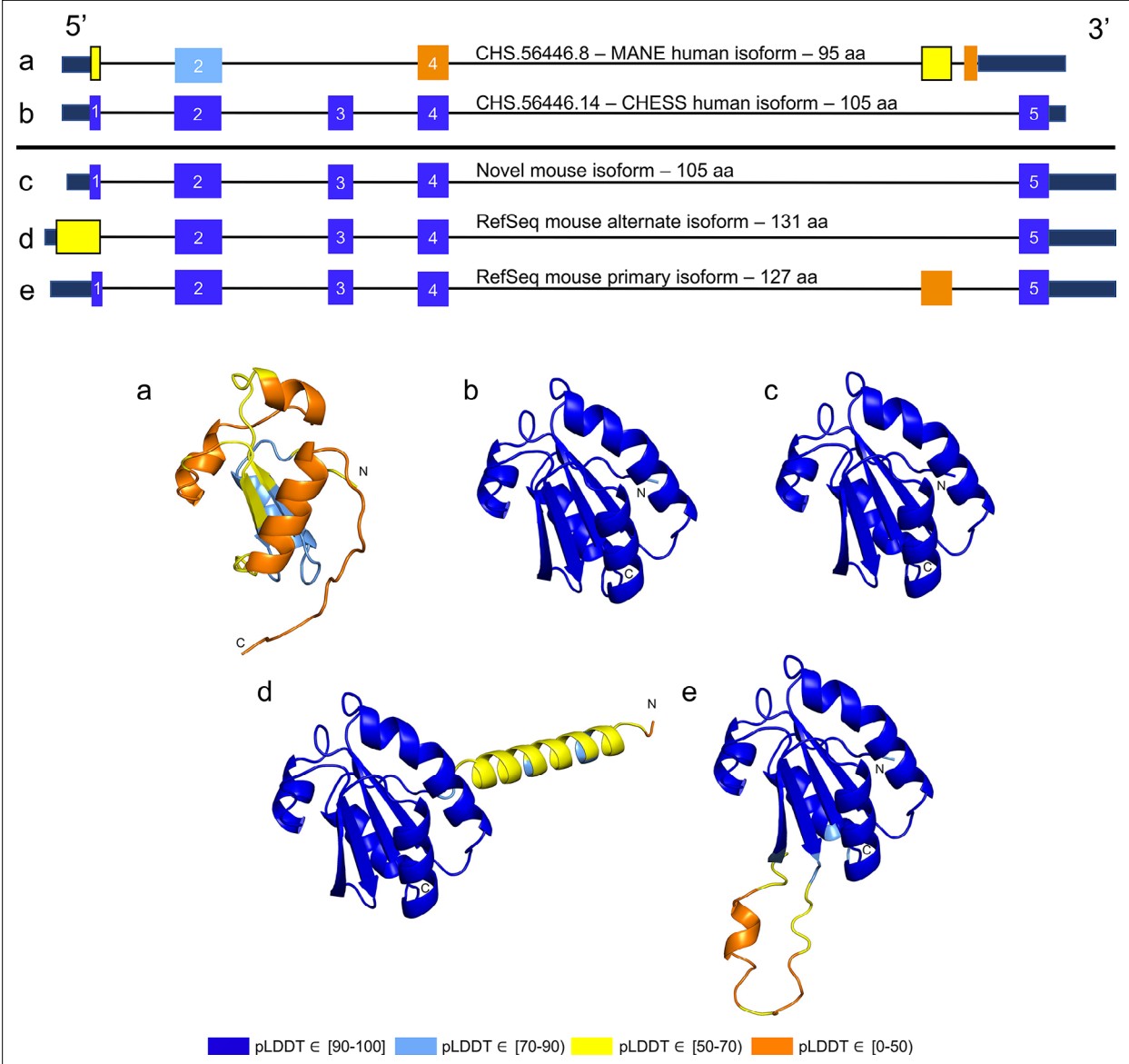

**Figure 9.** TXNDC8 human and mouse comparison. Intron-exon and predicted protein structure for TXNDC8 in human (**a and b**) and mouse (**c**, **d**, and **e**). Exons are colored according to their average predicted local distance difference test (pLDDT) score. The highest-scoring isoforms in both human (**b**) and mouse (**c**) share conserved intron-exon structure and nearly identical predicted protein structure.

## A novel protein-coding transcript in mouse

While examining the evolutionary conservation of the 3D structure of *TXNDC8* in human, we noticed that the predicted structure for the same gene in *Mus musculus* (house mouse) seemed to contain a poorly folded region similar to CHS.56446.6, a low-scoring human protein. Further inspection of *TXNDC8* in the mouse genome revealed that the primary RefSeq transcript contains a misfolding fifth exon, while the alternate RefSeq transcript skips the misfolding exon, similar to the functional human isoform, shown in *Figure 9*. Interestingly, both mouse transcripts contain a third exon homologous to the sequence missing in the human MANE isoform. A BLASTP search of the misfolding exon resulted in only a single significant hit outside the order *Rodentia* to an unnamed protein product. In contrast, a BLASTP search of the third exon present in all mouse transcripts, homologous to the missing exon in the human MANE transcript, revealed significant hits to at least 31 homologs outside *Rodentia*.

Unlike the functional human isoform, however, the RefSeq alternate mouse transcript is annotated with a start site 26 codons upstream of the translation initiation site annotated in the human

orthologs. A BLASTP search of the 26 amino acid additional sequence resulted in zero significant hits outside *Rodentia*. Folding this alternate mouse transcript revealed that these additional N-terminal amino acids fail to form any confident predicted protein structure. As a result, none of the three transcripts annotated in mouse fold into the highly conserved structure of *TXNDC8* in human. Predicted structures and exon alignments for isoforms in human and mouse are shown in *Figure 9*.

We hypothesized that a truly functional isoform of *TXNDC8* in mouse should be similar to the human isoform in 3D structure. Based on our observations, this similarity might be realized if the mouse alternate isoform simply started at the downstream start site that matches human, yielding a 105aa protein rather than the 131aa protein that is currently annotated. Folding the coding sequence of the alternate mouse transcript, minus the 26 N-terminal amino acid residues, revealed a predicted structure remarkably similar to both human and cattle *TXNDC8*. *Figure 5* shows the 3D alignment of the human and cow proteins to our predicted (105aa) isoform of mouse *TXNDC8*. Remarkably, the predicted average RMSD between aligned heavy atoms of the putative human and mouse proteins is just 0.83 Å. For reference, the atomic diameter of one carbon atom is 1.4 Å.

In a further investigation of mouse *TXNDC8* transcription, we aligned 8 gigabases of RNA-seq cDNA from a mouse testis sample (SRR18337982) to the GRCm38 reference genome using HISAT2 (*Kim et al., 2019*) then assembled transcripts using StringTie2 (*Kovaka et al., 2019*). This resulted in two putative transcripts at the mouse *TXNDC8* locus, with neither transcript containing the upstream start site present in the RefSeq annotation. Examination of the read coverage confirmed that both putative *TXNDC8* transcripts appear to use the start site of our proposed shorter protein-coding sequence, with 1060 reads supporting the canonical start site and zero reads supporting the upstream start site. As hypothesized, one of these newly assembled transcripts contained the protein-coding sequence necessary to exactly match our predicted structure-conserved isoform.

We believe this represents the first experimentally confirmed novel isoform in any organism discovered due to a hypothesis derived from comparison of computationally predicted protein structures. All in all, the structure-guided identification and subsequent experimental confirmation of a novel functional *TXNDC8* isoform in mouse demonstrates the potential of 3D protein structure prediction to enhance functional annotation in any genome.

## A resource for human annotation

The examples discussed here are only a small subset of the 130,700 unique protein structures we generated at 20,817 human gene loci. These structures and associated pLDDT scores have already been used to avoid filtering out transcripts encoding functional, clinically relevant isoforms while building the latest version of the CHESS human gene catalog. We provide all of these structures as a searchable and downloadable database, at isoform.io, to create a public resource for improving the annotation of the human genome. The large majority of CHESS isoforms in this collection have direct support from RNA-seq data, as they were assembled from the large GTEx collection, a high-quality set of deep RNA-seq experiments across dozens of human tissues. A very small number of MANE proteins were not assembled from GTEx data, but structures of these too are included in this analysis so that no MANE genes would be omitted.

In the online resource, we provide for each transcript: (1) the nucleotide and amino acid sequences; (2) the predicted structure, as a file that can be viewed in a standard structure viewer such as PyMOL (*Schrödinger, 2015*) (3) the pLDDT score of that structure; (4) the length of the isoform; (5) the number of GTEx samples in which the isoform was observed; (6) the maximum expression of the isoform in any tissue; (7) an indicator based on alignment of whether all introns are conserved in the mouse genome; (8) an interactive table with functionality to search and sort transcripts to find isoforms of interest; and (9) a Foldseek (*van Kempen et al., 2022*) interface to search any given protein structure against all 237,295 transcripts presented here. These predictions can be mined to discover, for example, cases where a known protein gets a surprisingly low pLDDT score, or where alternative isoforms have structures that get higher scores and appear more stable than previously reported forms of the same protein.

## Discussion

In this analysis, we demonstrated the ability to improve protein-coding gene annotation by predicting 3D protein structures. We searched tens of thousands of predicted structures of alternate isoforms of human genes and identified a subset that appear to fold more confidently than the isoforms found in MANE, a recently developed 'universal standard' for human gene annotation (*Morales et al., 2022*). We found hundreds of gene isoforms, all of which were supported by RNA-seq data, that outscored the corresponding MANE transcript. Inclusion of truly functional protein isoforms in future releases of human gene catalogs, particularly clinically focused catalogs such as MANE, may enable more accurate downstream analyses of these genes.

In the illustrative examples described here, we provide biological and evolutionary context for cases where an alternate human isoform appears clearly superior in structure to its canonical protein. Given the many additional high-scoring transcripts that we identified (*Supplementary file 3*), we expect further improvements in human annotation are yet to be discovered. More generally, we followed a structure-guided annotation strategy that may prove useful in refining the annotation of many non-human species as well.

We expect computational protein structure prediction to become an indispensable tool for future transcriptome annotation efforts. Still, the functionality of many proteins may not be revealed by structure prediction alone. Cases where substantial portions of a protein fail to form a stable structure, such as intrinsically disordered proteins, were not examined here. Additionally, non-functional isoforms may achieve a higher pLDDT simply due to the omission of small, yet functionally important intrinsically disordered regions. In these cases, it is necessary to contextualize results with both expression data, when available, and evolutionary sequence conservation analysis. If a low-scoring region is highly conserved across species or is consistently expressed, this still provides a strong indication of function regardless of foldability.

Although we restricted our analysis to whole-protein comparisons, comparing local structural portions of a protein, potentially near shape-sensitive ligand binding sites (*Greer et al., 1994*), may enable similar analysis in these proteins. Further advances in predicting structures for multi-chain protein complexes (*Evans et al., 2022*), as well as improvements in prediction efficiency in large proteins, may expand the range of genes that may be analyzed. An important caveat is that in some cases, truly functional isoforms may receive low predicted folding scores relative to well-folded functional alternate isoforms within the same gene. Thus, structure prediction alone is not always sufficient to make functional claims about any individual protein isoform.

### Ideas and speculation

Though the complete sequence of the human genome has been revealed (*Nurk et al., 2022*), the annotation of the human genome, by far the most comprehensively studied, remains far from finished. The use of accurate predicted protein structures for gene annotation, as we have done here, represents a new paradigm, not only for human gene discovery but for all other species as well. For decades, the scientific community has relied principally on two methods to discover and validate protein-coding genes at the genome scale: sequencing of transcripts (or cDNAs), and alignment of DNA and protein sequences to detect evolutionary conservation in other species. Protein structure holds valuable information regarding biological functionality, providing an independent and powerful tool to complement these methods. The analysis described here takes a first step toward improving genome annotation of humans using structure prediction, but specific methods deploying this powerful new tool on a broader scale will require fundamentally new computational strategies. As such, development of comprehensive genome annotation protocols incorporating protein structure prediction will remain an area of active investigation for years to come.

## Materials and methods

### Protein structure prediction

We folded all transcripts in the CHESS annotation less than 1000 amino acids in length. Similar to the initial effort to fold the human proteome (*Tunyasuvunakool et al., 2021*), the length limit was chosen to make the overall computational runtime feasible. This yielded 233,973 transcripts representing 127,398 unique protein-coding sequences at 20,666 loci. Coding sequences for CHESS transcripts

were determined with ORFanage (*Varabyou et al., 2021*). For each protein sequence, we generated a multiple sequence alignment by aligning them with ColabFold's MMseqs2 (*Steinegger and Söding, 2017*) workflow (colabfold_search) against the UniRef100 (*Suzek et al., 2015*) (2021/03) and Colab-FoldDB (2021/08) database. Structure predictions were made with ColabFold (commit 3398d3) using AlphaFold2 and MMseqs2 version 13.45111. To speed up the search, we set the sensitivity setting to 7 (-s 7). We predicted each structure using colabfold_batch and stopped the process early if a pLDDT of at least 85 was reached by any model (--stop-at-score 85) or if a model produced a pLDDT less than 70 (--stop-at-score-below 70). All models were ranked by pLDDT in descending order. Runtime was estimated from a sample of 500 proteins randomly selected from the 127,398 structures. Prediction of all structures on 8 × A5000 GPUs required 34 days. Multiple sequence alignment took 34 hr using MMseqs2 on an AMD EPYC 7742 CPU with 64 cores.

## Filtering MANE comparisons

To generate the 940 protein isoforms in *Supplementary file 3*, we used the following filtering criteria and procedures. For each isoform with a distinct coding sequence located at a MANE v1.0 locus and with coding sequence overlapping a MANE annotated protein, we compared the pLDDT score to that of the associated MANE protein. As described previously, pLDDT is a reliable measure of the confidence in a structure, where predictions with $70 \leq pLDDT \leq 90$ are confident, those with pLDDT >90 are highly confident, and those below 50 represent low-confidence structures and may be disordered proteins (*Varadi et al., 2022*). We only considered alternative isoforms that had a pLDDT score ≥70, indicating a generally well-folded protein, to avoid including any intrinsically disordered proteins (*Ruff and Pappu, 2021*). Filtering and general analysis was performed in Colab (https://colab.research. google.com) with Python version 3.8.15.

We selected isoforms that, when compared to the MANE isoform for the same gene, scored at least 5% higher as measured by pLDDT and were at least 90% as long. Additionally, to capture cases where the MANE transcript might be missing functional sequence elements, we selected alternate isoforms that were at least 5% longer than the MANE isoform, that had equal or higher pLDDT scores, and that were assembled in an equal or higher number of GTEx samples. Finally, to capture cases where an alternate isoform might be functional despite being substantially shorter than the MANE protein, we selected isoforms at least 50% as long as the MANE protein where the alternate isoform scored at least 5% higher and was assembled in an equal or higher number of GTEx samples. After applying these filters, we observed that in some cases, a processed pseudogene (*Zhang et al., 2002*) contained within an intron outscored the associated primary transcript. To eliminate such cases, we used GFFcompare (*Pertea and Pertea, 2020*) to ensure that isoforms overlapped their MANE transcript's coding sequence. When multiple alternate isoforms contained the same coding sequence, and thus received the same pLDDT score, we selected the isoform assembled in the highest number of GTEx samples.

## Annotation sources

Transcript annotations for MANE, CHESS, RefSeq, and GENCODE were retrieved from the following sources. The MANE v1.0 database was downloaded from NCBI at https://www.ncbi.nlm.nih.gov/ refseq/MANE. Annotations from the CHESS v3.0 database were retrieved from http://ccb.jhu.edu/ chess. Additional CHESS annotations came from an unpublished set of transcript assemblies created as part of the process of building CHESS v3.0; these were assembled from approximately 10,000 GTEx RNA-seq experiments across 31 tissues using StringTie2 (*Kovaka et al., 2019*). Transcripts from this set were given a CHESS ID starting in 'hypothetical' if the locus was missing from CHESS v3.0, or else given an ID ending in 'altN' if the locus was present in CHESS v3.0 but the exact isoform was not. Note that many of these, particularly those with a poor protein folding score, were not retained in the final CHESS v3.0 database. RefSeq annotations (releases 109 and 110) were downloaded from https:// www.ncbi.nlm.nih.gov/projects/genome/guide/human/index.shtml. GENCODE (v38, v39, v40) annotations were collected from https://www.gencodegenes.org/human/.

## Visualization and atomic alignment

All visualizations and 3D protein structure atomic alignments were performed in PyMol (*Schrödinger, 2015*) version 2.5.2 using non-orthoscopic view, white background, and ray trace 1200,1200. RMSD

were calculated without excluding any outliers. Ramachandran plots were created using PyRAMA version 2.0.2 with Richardson (*Lovell et al., 2003*) standard psi and phi values for all amino acids excluding glycine and proline. Intron-exon structure plots were produced with MISO (*Katz et al., 2010*) (commit b714021) and TieBrush (*Varabyou et al., 2021*) (commit e986d64).

### RNA-seq quantification of the human ASMT gene

RNA-seq data were downloaded from NCBI for run SRR5756467 from BioSample SAMN07278516, a pineal gland from a patient who died at midnight. A detailed summary of the experimental protocol used to generate these data can be found in NCBI BioProject PRJNA391921. Isoform-level quantification was performed using Salmon (*Patro et al., 2017*) version 1.8.0.

### RNA-seq assembly of mouse TXNDC8

RNA-seq data were downloaded for run SRR18337982 from BioSample SAMN26725167, a tissue sample from the testis of a control mouse. A detailed summary of the experimental protocol used to generate these data can be found in NCBI BioProject PRJNA816862. cDNA reads were aligned to the mm39 reference genome using HISAT2 (*Kim et al., 2019*) version 2.1.0 then assembled into transcripts using StringTie2 (*Kovaka et al., 2019*) version 2.2.1.

### Intron conservation in human and mouse

We assessed the conservation of GT-AG intron boundaries between CHESS human transcripts and transcripts from the GRCm38 mouse reference genome. Data for the human-mouse alignment was extracted from a 30-species alignment anchored on GRCh38 that was downloaded from the UCSC genome browser (*Navarro Gonzalez et al., 2021*). We used MafIO in BioPython (*Cock et al., 2009*) version 1.71 to check if all intron boundaries were conserved between mouse and human transcripts. In the supplementary files, a value of 'TRUE' in the 'introns in mouse' column indicates that all boundaries were conserved, while 'FALSE' means that at least one splice site (either a GT at a donor site or an AG at an acceptor site) was not conserved in the alignment.

## Acknowledgements

The authors would like to thank all members of the Salzberg, Pertea, and Steinegger labs, as well as David J Lipman for helpful feedback during project conceptualization, Benjamin Langmead for publicly hosting bulk data files, and Do-Yoon Kim for creating the isoform.io logo.

## Additional information

### Funding

| Funder | Grant reference number | Author |
| --- | --- | --- |
| National Institutes of Health | R01-HG006677 | Steven L Salzberg |
| National Institutes of Health | R35-GM130151 | Steven L Salzberg |
| National Research Foundation of Korea | 2019R1-A6A1-A10073437 | Martin Steinegger |
| National Research Foundation of Korea | 2020M3-A9G7-103933 | Martin Steinegger |
| National Research Foundation of Korea | 2021-R1C1-C102065 | Martin Steinegger |
| National Research Foundation of Korea | 2021-M3A9-I4021220 | Martin Steinegger |
| Seoul National University | Creative-Pioneering Researchers Program | Martin Steinegger |

| Funder | Grant reference number | Author |
|---|---|---|

The funders had no role in study design, data collection and interpretation, or the decision to submit the work for publication.

## Author contributions

Markus J Sommer, Conceptualization, Data curation, Software, Formal analysis, Validation, Investigation, Visualization, Methodology, Writing – original draft, Project administration, Writing – review and editing; Sooyoung Cha, Sukhwan Park, Ilia Minkin, Data curation, Software, Investigation; Ales Varabyou, Data curation, Software, Visualization; Natalia Rincon, Visualization; Mihaela Pertea, Funding acquisition, Validation, Investigation, Methodology, Writing – review and editing; Martin Steinegger, Conceptualization, Resources, Supervision, Funding acquisition, Methodology, Writing – review and editing; Steven L Salzberg, Conceptualization, Resources, Formal analysis, Supervision, Funding acquisition, Validation, Investigation, Methodology, Writing – original draft, Project administration, Writing – review and editing

## Author ORCIDs

Markus J Sommer  http://orcid.org/0000-0003-3414-1875
Sooyoung Cha  http://orcid.org/0000-0001-7211-4603
Steven L Salzberg  http://orcid.org/0000-0002-8859-7432

## Decision letter and Author response

Decision letter https://doi.org/10.7554/eLife.82556.sa1
Author response https://doi.org/10.7554/eLife.82556.sa2

# Additional files

## Supplementary files

• Supplementary file 1. All isoform summary. Folding scores from ColabFold for each transcript from a preliminary new build of the Comprehensive Human Expressed SequenceS (CHESS) database that contained a protein-coding sequence (CDS) that was under 1000aa in length. For transcripts already contained in the released CHESS v3.0 database, the identifier from that database is provided. If the transcript maps to a known gene locus X but is a novel isoform, it is shown with the identifier CHS.X.altY. If a transcript occurs at a novel locus X, the identifier is hypothetical.X.Y, where Y identifies the isoform number. Additional columns show the gene name, the RefSeq ID (release 110), the GENCODE ID (release 40), the predicted local distance difference test (pLDDT) (folding) score, and a flag indicating whether all intron boundaries (for multi-exon genes) are conserved in the mouse genome.

• Supplementary file 2. Matched Annotation from NCBI and EMBL-EBI (MANE) comparison summary. Folding scores and additional data for all Comprehensive Human Expressed SequenceS (CHESS) transcripts that match genes in the MANE v1.0 dataset, limited to protein sequences under 1000aa in length. Transcripts must overlap the annotated CDS of the MANE transcript to be included. Columns include: *CHESS_ID_isoform*, the CHESS identifier of the alternate isoform transcript; *CHESS_ID_MANE*, the CHESS identifier of the MANE transcript at the same locus; *gene*, the gene name; *aa_length_isoform*, the amino acid length of the alternate isoform's CDS; *aa_length_MANE*, the amino acid length of the MANE transcript's CDS; *length_ratio*, the ratio of the alternate isoform length to the MANE isoform length; *pLDDT_isoform*, the predicted folding score of the alternate isoform; *pLDDT_MANE*, the predicted folding score of the MANE isoform; *pLDDT_ratio*, the ratio of the alternate isoform folding score to the MANE isoform folding score; *GTEx_samples_observed_isoform*, the total number of GTEx samples where the alternate isoform was observed at least once; *GTEx_samples_observed_MANE,* the total number of GTEx samples where the MANE isoform was observed at least once; *GTEx_top_tissue_name_isoform*, the name of the tissue in which the alternate isoform was observed in the highest number of samples; *GTEx_top_tissue_name_MANE*, the name of the tissue in which the MANE isoform was observed in the highest number of samples; *GTEx_top_tissue_TPM_isoform*, the average TPM of the alternate isoform in the named tissue; *GTEx_top_tissue_TPM_MANE*, the observed transcripts per million (TPM) of the MANE isoform in the named tissue; *introns_conserved_in_mouse_isoform*, an indicator of whether introns are conserved between the alternate human isoform and any annotated isoform in the GRCm38 mouse reference genome; *introns_conserved_in_mouse_MANE*, an indicator of

whether introns are conserved between the MANE human isoform and any annotated isoform in the GRCm38 mouse reference genome.

• Supplementary file 3. Matched Annotation from NCBI and EMBL-EBI (MANE) comparison summary, filtered subset. A filtered set of Comprehensive Human Expressed SequenceS (CHESS) transcripts compared to MANE according to the criteria detailed in the 'Filtering MANE comparisons' section of the Materials and methods. Uses the same column names as *Supplementary file 2*.

• MDAR checklist

### Data availability

Gene identifiers for all predicted protein isoforms as well as pLDDT scores and evolutionary conservation data from mouse can be found in Supplementary file 1. Predicted scores and GTEx expression data for all isoforms overlapping a MANE locus can be found in Supplementary file 2. Data for the 940 alternate isoforms with evidence of relatively superior structure, and possibly superior function, can be found in Supplementary file 3. Additionally, all data can be downloaded from Figshare (https://doi. org/10.6084/m9.figshare.21802476.v1).

The following dataset was generated:

| Author(s) | Year | Dataset title | Dataset URL | Database and Identifier |
|---|---|---|---|---|
| Sommer M, Cha S, Varabyou A, Rincon N, Park S, Minkin I, Pertea M, Steinegger M, Salzberg S | 2023 | Structure-guided isoform identification for the human transcriptome | https://doi.org/10. 6084/m9.figshare. 21802476.v1 | Figshare, 10.6084/ m9.figshare.21802476.v1 |

The following previously published datasets were used:

| Author(s) | Year | Dataset title | Dataset URL | Database and Identifier |
|---|---|---|---|---|
| Pertea M, Salzberg S | 2018 | CHESS 3.0 | http://ccb.jhu.edu/ chess/ | CHESS, 2.2 |
| GTEx Consortium | 2013 | The Genotype-Tissue Expression (GTEx) project | https://gtexportal. org/home/ | GTEx, V8 |
| Morales J, Pujar S | 2022 | Matched Annotation from NCBI and EMBL-EBI (MANE) | https://www.ncbi. nlm.nih.gov/refseq/ MANE/ | NCBI RefSeq, MANE |
| Chang E | 2020 | A multi-species multi-timepoint transcriptome database and webpage for the pineal gland and retina | https://www.ncbi. nlm.nih.gov/sra/ SRR5756467 | NCBI Sequence Read Archive, SRR5756467 |
| Wang et al. | 2022 | Tes45; *Mus musculus* | https://www.ncbi.nlm. nih.gov/sra/20655315/ SRR18337982 | NCBI Sequence Read Archive, SRR18337982 |

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
