## [Editor Report]

This study applies AlphaFold to the CHESS selection of transcripts with the goal of generating predicted 3D protein structures and a quality measure of folding, the pLDDT score. From these data, the authors build a database for result exploration, documented by several examples, including proteins, where the authors propose the pLDDT score as a measure of presumed superior biological functionality over other isoforms. These results will be highly relevant for anyone working with proteins that occur in different isoforms.

---

## [Decision Letter]

**Decision letter after peer review:**

Thank you for submitting your article "Structure-guided isoform identification for the human transcriptome" for consideration by *eLife*. Your article has been reviewed by 2 peer reviewers, and the evaluation has been overseen by a Reviewing Editor and Volker Dötsch as the Senior Editor. The following individual involved in review of your submission has agreed to reveal their identity: Anna Poetsch (Reviewer #2).

Essential revisions:

1) Using the representative isoform from the MANE database the authors provide 5 exemplar genes whose MANE representative isoform when translated yields a protein structure with a lower pLDDT score than a non-representative isoform, suggesting that the protein stability of the representative isoform is suspect. The authors clearly explain that not all isoforms encode a well folded functional protein. Thus, it would be helpful for the authors to provide a clearer indication of how common the non-exemplar representative isoforms in the MANE database may have a lower pLDDT score. Armed with this a false positive and negative estimate could be provided.

2) The authors evaluate the single representative isoform selected for each human protein coding gene in the MANE database. In addition, the authors provide a list of isoforms with the highest pLDDT score for each human gene. However, it would be useful for the authors to provide a section that speaks to the confounding conditions and sequence features are understood to be confounding in evaluating the use of pLDDT scores. For example, since evolutionary conservation and synteny considerations of isoform sequences are an important consideration in evaluating the potential utility of an isoform, what would the authors recommend if the pLDDT scores are elevated but one or more of the other considerations are less than ideal?

3) The study is based on the elegant idea to aid genome annotation through 3D structure prediction. This is a very powerful approach that allows large-scale data generation for functional interpretation. This approach appears technically sound and well executed (although I may miss details not being a protein expert). However, in my opinion, the authors could make more use of the potential of their approach. From the big-data start, they seem to directly restrict themselves to interesting examples. I am missing a global analysis that shows the bigger picture of their results. Given that they have generated structures from 90,415 isoforms, each associated with a pLDDT score, conservation scores, length, expression levels and other quantifiable data listed on page 18. I would wish for a comprehensive analysis of these data and their potential before applying the focus on a few (admittedly very nice) examples.

4) One of the weak spots of such an analysis is the relationship between foldability and functional relevance. Disordered regions would imply reduced relevance due to poor pLDDT scores, which may be a misleading conclusion. While this may be a problem difficult to solve with this approach, it still needs to be addressed and discussed throughout the paper and particularly as part of the global analysis, not just in the context of examples.

*Reviewer #1 (Recommendations for the authors):*

The manuscript is well written and speaks to an important and timely issue concerning the number how to annotate and evaluate isoforms for each gene. As the number of isoforms for a genome continues to increase it will be helpful to provide some logic to distinguish among the isoforms.

Using the representative isoform from the MANE database the authors provide 5 exemplar genes whose MANE representative isoform when translated yields a protein structure with a lower pLDDT score than a non-representative isoform, suggesting that the protein stability of the representative isoform is suspect. The authors clearly explain that not all isoforms encode a well folded functional protein. Thus, it would be helpful for the authors to provide a clearer indication of how common the non-exemplar representative isoforms in the MANE database may have a lower pLDDT score. Armed with this a false positive and negative estimate could be provided.

The authors evaluate the single representative isoform selected for each human protein coding gene in the MANE database. In addition, the authors provide a list of isoforms with the highest pLDDT score for each human gene. However, it would be useful for the authors to provide a section that speaks to the confounding conditions and sequence features are understood to be confounding in evaluating the use of pLDDT scores. For example, since evolutionary conservation and synteny considerations of isoform sequences are important consideration in evaluating the potential utility of an isoform, what would the authors recommend if the pLDDT scores are elevated but one or more of the other considerations are less than ideal?

Finally, the authors are encouraged to provide readers with a reasoned argument that the addition of structure-guided isoform considerations is more than an incremental advancement in the annotation of the human transcriptome.

---

## [Author Response]

Essential revisions:1) Using the representative isoform from the MANE database the authors provide 5 exemplar genes whose MANE representative isoform when translated yields a protein structure with a lower pLDDT score than a non-representative isoform, suggesting that the protein stability of the representative isoform is suspect. The authors clearly explain that not all isoforms encode a well folded functional protein. Thus, it would be helpful for the authors to provide a clearer indication of how common the non-exemplar representative isoforms in the MANE database may have a lower pLDDT score. Armed with this a false positive and negative estimate could be provided.

We have updated our analysis to include all CHESS 3 isoforms <= 1000aa (rather than the previous limit of 500aa) and have lifted over all CHESS 3 matching predictions from the AlphaFold Protein Structure Database, which increases our maximum length limit to 2699aa. This resulted in >98% of all human loci containing at least one protein structure prediction, which we hope makes the following additions to the global analysis more truly global.

We have rewritten the “Scoring the transcriptome” section of the results to include a more comprehensive explanation of the comparison between MANE and the higher-scoring isoforms. To clarify how commonly the non-exemplar isoforms (i.e., the ones not included in MANE) outscore the canonical isoform, we added a count of the number of isoforms that score a higher pLDDT vs. their associated MANE isoform. However, even with RNA-seq, protein structure prediction, and evolutionary conservation evidence, we do not believe we can create a gold standard for which transcripts are truly functional, and thus we cannot provide an unbiased estimate of false positives and negatives for MANE. Many of the hundreds of isoforms identified in Table S3 represent poorly-studied proteins where the additional experimental evidence used to further analyze our Exemplary Predictions simply does not exist. We believe that our addition of more explicit numbers, as now provided in the updated Results section, provides a fair comparison to MANE despite lacking formal false positive and negative estimates.

2) The authors evaluate the single representative isoform selected for each human protein coding gene in the MANE database. In addition, the authors provide a list of isoforms with the highest pLDDT score for each human gene. However, it would be useful for the authors to provide a section that speaks to the confounding conditions and sequence features are understood to be confounding in evaluating the use of pLDDT scores. For example, since evolutionary conservation and synteny considerations of isoform sequences are an important consideration in evaluating the potential utility of an isoform, what would the authors recommend if the pLDDT scores are elevated but one or more of the other considerations are less than ideal?

We have added a more thorough explanation to the “Scoring the transcriptome” section on why certain isoforms may achieve a high pLDDT but still be non-functional. Due in part to the problem of short protein fragments (now mentioned in the results), which sometimes get misleadingly high scores, our set of filtered transcripts is based on a combination of foldability and RNA-seq expression evidence rather than foldability alone. We hope this clearer description of the reasoning behind our methods, alongside the examples which incorporate additional experimental evidence, may show future researchers how to handle cases where considerations are less than ideal, e.g. ASMT lacking expression data in GTEx because it is expressed primarily in the pineal gland which was not sampled. Our overall goal here is not to provide a definitive solution to the problem of determining which isoforms are functional (as we believe any attempt to do so would be suboptimal and incomplete at this time), but rather to provide a type of field-guide and associated genome-wide database which can be used as resource in future efforts.

3) The study is based on the elegant idea to aid genome annotation through 3D structure prediction. This is a very powerful approach that allows large-scale data generation for functional interpretation. This approach appears technically sound and well executed (although I may miss details not being a protein expert). However, in my opinion, the authors could make more use of the potential of their approach. From the big-data start, they seem to directly restrict themselves to interesting examples. I am missing a global analysis that shows the bigger picture of their results. Given that they have generated structures from 90,415 isoforms, each associated with a pLDDT score, conservation scores, length, expression levels and other quantifiable data listed on page 18. I would wish for a comprehensive analysis of these data and their potential before applying the focus on a few (admittedly very nice) examples.

We have added Figure 1, a transcriptome-wide view of the relationship between pLDDT vs. protein length and GTEx expression to the results. The figures shows a lack of a clear linear relationship between pLDDT and length or expression, which implies that protein structure prediction provides an orthogonal source of useful information for annotation. This is also noted in the updated manuscript.

4) One of the weak spots of such an analysis is the relationship between foldability and functional relevance. Disordered regions would imply reduced relevance due to poor pLDDT scores, which may be a misleading conclusion. While this may be a problem difficult to solve with this approach, it still needs to be addressed and discussed throughout the paper and particularly as part of the global analysis, not just in the context of examples.

This is a great point. Many functional proteins are either entirely disordered or contain intrinsically disordered regions, and protein folding will inherently fail to provide useful information in these cases. We have added a section as part of the global analysis in the results which quantifies how often we observe this in our data. Additionally, we have expanded our Discussion section to include a more thorough description of how we believe this problem should be approached by researchers looking at protein foldability in context with expression and evolutionary evidence.